# Coping Mechanisms during the War in Ukraine: A Cross-Sectional Assessment among Romanian Population

**DOI:** 10.3390/healthcare11101412

**Published:** 2023-05-13

**Authors:** Cătălina Angela Crișan, Zaki Milhem, Roland Stretea, Radu Mihai Hossu, Ionuț Stelian Florean, Răzvan Mircea Cherecheș

**Affiliations:** 1Department of Neurosciences, Psychiatry and Pediatric Psychiatry, Faculty of Medicine, Iuliu Hațieganu University of Medicine and Pharmacy, 400349 Cluj-Napoca, Romania; 2Clinical Hospital of Infectious Diseases, 400347 Cluj-Napoca, Romania; 3Internet Retailing Office SRL, 500209 Brașov, Romania; 4Department of Clinical Psychology and Psychotherapy, Babeş-Bolyai University, 400084 Cluj-Napoca, Romania; 5RoNeuro Institute, Center for Research and Diagnosis of Neurological Diseases, 400012 Cluj-Napoca, Romania; 6Department of Public Health, College of Political, Administrative and Communication Sciences, Babeș-Bolyai University, 400591 Cluj-Napoca, Romania

**Keywords:** war, mental health, coping mechanisms, quality of life

## Abstract

On 24 February 2022, the Russian Federation invaded Ukraine, starting a military conflict that soon turned into a full-scale war. The Romanians have been actively involved in helping their neighbors, causing the accumulation of emotional and mental pressure upon the Romanian population exposed to such a close military conflict. This cross-sectional study assessed, through an online survey (1586 adult Romanian residents), the primary coping mechanisms, quality of life and anxiety levels in response to the psychological trauma associated with the Russo—Ukrainian war. Based on the results, focusing on and venting emotions along with behavioral disengagement were the coping strategies that had the strongest negative impact on anxiety and well-being. On the other hand, positive reinterpretation and growth were associated with less anxiety, a higher degree of overall health and better quality of life. To the best of our knowledge, this study is the first of its kind to be performed among the Romanian population so far. Thus, we equip mental health practitioners with the tools (real-life evidence data) that will allow them to establish a more meaningful doctor—patient relationship, maximizing therapy results.

## 1. Introduction

Because of the 2008 war in Georgia, the 2014 annexation of Crimea, the ongoing conflicts in Donbas and the massive military buildup of Russia in the fall of 2021, the actual invasion should not have come as a surprise. However, when Russia invaded Ukraine on 24 February 2022, the whole world was shocked [1].

Since the war started, the hostility and brutality against the Ukrainians have led to disastrous consequences, including loss of life, severe injuries and mass population movements. In fact, according to the latest reports available, over 8000 people have been killed, 13,000 injured, 5.4M internally displaced and 8M made refugees in European countries. Moreover, the incidence of some infectious diseases, including AIDS, tuberculosis and COVID-19, has also increased. About 18M Ukrainians have been in need of humanitarian aid so far [2,3], while being exposed to a great deal of psychological trauma during this war [4,5].

The Russo—Ukrainian war, which succeeded the COVID-19 pandemic, is continuing to affect Europe’s already fragile socio-economic sector. Experiencing close military conflict has not been easy for the neighboring countries. The large refugee movements, air surveillance, nuclear war anxiety and negative economic impact have all contributed to the already upscaling fear of the war next door. This has increased the spectrum of emotional and cognitive distress, with a severe impact on citizens’ mental health, especially since the effects of this full-scale war were already visible in their everyday lives. Depression, anxiety and post-traumatic stress were all found to be elevated in those affected by the nearby war. Countries such as Poland, Romania, the Republic of Moldova, Hungary, Slovakia, Bulgaria and the Czech Republic have been particularly affected [6,7,8,9,10,11].

Romania is the European Union (EU) country that shares the longest border with Ukraine (Figure 1) and, since the beginning of the Russo—Ukrainian war, Romanians have been actively involved in helping their Ukrainian neighbors [12]. Their proactive mobilization included several public protests against the Russian invasion, fundraising actions, social media engagement to increase awareness, transportation of essential supplies across the border, as well as providing shelter, access to health services, education and jobs for the Ukrainian refugees [13]. However, this prosocial behavior comes with a cost. Several studies, including those of Jawaid et al. and Haisan et al., highlight the risk of those involved in major humanitarian actions in terms of increased mental pressure due to their exposure to others’ trauma [11,14,15].

As an EU and North Atlantic Treaty Organization (NATO) member state, Romania has had to reinforce its responsibilities as a geopolitical supporter of Ukraine and its war refugees. Besides the social effects of living in close proximity to a military conflict, there is a great deal of psychological burden caused by the economic consequences of this war. Alongside the expenses associated with the integration of Ukrainian refugees [16], different industry problems, businesses closing, job losses, rising inflation, financial instability and the energy crisis are also causing economic pressure. Altogether, these negative economic changes and financial insecurities elevate the psychological pressure on the Romanian population [17,18,19].

Coping is described as a repertoire of subconscious thoughts or unconscious adaptive behavior that helps the human subject to manage emotional pressure and reduce or tolerate stress. These internal “defense mechanisms”, generally called coping styles, are a set of relatively stable traits that determine the individual’s behavior in response to internal or external stress [20,21,22]. Understanding coping mechanisms is crucial to helping medical doctors choose the best approach to build an effective relationship with their patients [23]. In this regard, several coping scales were developed and proved reliable for measuring the type of coping mechanism a person exhibits [24]. A small body of studies has assessed the coping mechanisms of Ukrainians during this time of war. Xu et al. evaluated the mental health status and the coping and resilience behavior of 14,000 Ukrainians, aged 18 or older, during the initial period of the Russian invasion. Their findings indicated an elevated rate of psychological distress, anxiety, depression and insomnia, whereas the most prevalent productive coping strategies were using instrumental support, behavioral disengagement, self-distraction and planning [25].

This cross-sectional study was designed to assess the primary coping mechanisms, quality of life and anxiety levels in response to the psychological trauma associated with the Russo—Ukrainian war among the general population in Romania. To the best of our knowledge, evaluating the coping mechanisms of Romanians during this very close military conflict is the first study of its kind and provides valuable information that could aid doctor—patient relationships, ensuring better management of post-traumatic symptoms.

## 2. Materials and Methods

### 2.1. Participants

A comprehensive study cohort took part in this investigation (*n* = 1586), including 1278 females and 308 males. All participants were 18 years of age or above (188 were 18–25 years old, 147 were 26–30 years old, 506 were 31–40 years old, 466 were 41–50 years old, 244 were 51–60 years old, 65 were 61–70 years old and 12 were older than 71) and had permanent residency in Romania at the time of completing the online survey. Most participants lived in urban areas (*n* = 1400), while just a small number of the respondents was from a rural area (*n* = 186). Most participants had a bachelor’s degree (*n* = 674), postgraduate studies (*n* = 635) or were high-school graduates (*n* = 254). Most of the participants were married (*n* = 878), while the others were unmarried (*n* = 370), divorced (*n* = 124), cohabitating (*n* = 161) and widowed (*n* = 28). Regarding professional status, most of the participants had a job (*n* = 1062), while the remaining respondents were entrepreneurs (*n* = 132), freelancers (*n* = 162), retired (*n* = 80), or students (*n* = 77).

### 2.2. Procedure

A cross-sectional study was conducted by our group of researchers from the “Iuliu Hatieganu” University of Medicine and Pharmacy Cluj-Napoca, Romania. As such, between March 2022 and May 2022, we evaluated the participants’ anxiety levels, their quality of life and their coping mechanisms during the Russo—Ukrainian War. A one-time Google-Forms-based anonymous online survey was mass disseminated online through social media platforms (Facebook, Instagram, Twitter) and via emails. The link was also shared on the university website, in online publications and promoted through a televised interview. A copy of this questionnaire and an Excel spreadsheet containing all participants’ responses can be accessed in the Appendix A section.

This anonymous online questionnaire had a completion time of approximately 30 min, comprised 5 parts, 112 items and was designed to assess multiple parameters defining quality of life, such as anxiety symptoms, physical health, psychological health, social relationships and environment, but also variables describing different types of coping. A more detailed description of our survey is provided in the 2.3. Measures chapter. In order to maximize the study cohort size, the only inclusion criteria used were residency and age. In this regard, only permanent Romanian residents over 18 years of age were selected for the study. The only exclusion criterion was a refusal to give informed consent (if all questions were otherwise answered). One of the respondents recorded their completion of the form, but did not give their informed consent and, as such, they were excluded from the study. All other respondents that completed the form also gave their informed consent at the beginning and, as such, all recorded surveys were eligible to be taken into consideration. Google Forms does not record participants that do not complete the survey (for example, if one quits the survey mid-completion) and, as such, there is no way to know the total number of people that accessed our link. Therefore, out of 1587 respondents, 1586 gave informed consent and completed the survey thus being eligible to further take part in our study.

Each participant received a written explanation of the research and gave their informed consent to take part in the study. All data were securely held in confidentiality according to the General Data Protection Regulation (GDPR) agreement. This study was approved by the Ethical Committee of the “Iuliu Hatieganu” University of Medicine and Pharmacy Cluj-Napoca, Romania (AVZ84/29.03.2022). Respondents’ confidentiality will be maintained at all times. The findings will be widely disseminated in peer-reviewed journals, at conferences, through user networks and to policymakers and relevant clinical groups.

### 2.3. Measures

In the first section of the online survey, prospective participants were presented with written terms and conditions, which they had to approve to continue in the study. The purpose and duration of the study, the voluntary character of their participation, the possibility to withdraw consent at any time and a GDPR statement regarding data confidentiality were explained in detail.

The next section of the survey was composed of specific questions designed to collect the most relevant socio-demographic features, such as age, gender, education level and marital and professional status. Participants were also asked about substance use during wartime: smoking habits (whether they started or stopped smoking, potential increase or decrease in smoking), alcohol consumption (number of drinks/week) and whether or not they used psychoactive drugs. Finally, they were asked about whether they accessed medical services for any psychiatric disorders and whether they had an acute episode of a preexisting psychiatric disorder during wartime.

Quality of life was measured using the World Health Organization Quality-of-Life Scale—Short Form (WHOQOL-BREF), validated across cultures, accessible in Romanian. This form is a tool for measuring the quality of life, which captures various subjective factors, such as physical and mental well-being, level of independence, social connections and interactions with key elements of the environment. The scale has 27 questions, each with scores from 1 to 5, divided into four subscales corresponding to the aforementioned factors: mental area, physical area, environmental area and social relationships area. As the score gets higher in each individual subscale, the quality of life in that respective area increases [26]. The scales had good reliability as indicated by Cronbach’s alpha: physical health (0.64), psychological health (0.61) social relationships (0.70) and environment (0.75).

Coping mechanisms were measured using the Romanian version of the COPE (Coping Orientation to Problems Experienced) inventory, a multidimensional form that assesses the methods used by people to respond to stress. It comprises fifteen 4-item scales grouped into three categories: problem-focused methods (active coping, restraint, use of instrumental support, planning and suppression of competing activities), emotion-focused methods (use of emotional support, humor, positive reinterpretation, acceptance and turning to religion) and dysfunctional methods (venting, substance use, mental disengagement, behavioral disengagement and denial). Each scale has 4 items that frame the respective coping method into personal options and ways of action (for example, active coping—“I take additional action to try to get rid of the problem” or acceptance—“I learn to live with it”). Subjects were asked “How often did you do this when you experienced the stressful event”; with responses ranging on a 4-point scale from “did not do this at all” to “did this a lot” [27]. Regarding the reliability of the instrument, we computed Cronbach’s alpha for each subscale. We found acceptable values for positive reinterpretation and growth (0.67), focus on and venting emotions (0.69), use of instrumental social support (0.75), active coping (0.68), denial (0.71), religious coping (0.90), humor (0.94), behavioral disengagement (0.76), restraint (0.55), use of emotional social support (0.86), substance use (0.96), acceptance (0.76), suppression of competing activities (0.65) and planning (.81). However, the reliability for mental disengagement was deemed unacceptable (alpha = 0.49) and this subscale was excluded from all subsequent analyses.

Anxiety levels were measured using the Romanian version of the Hamilton Anxiety Rating Scale (HAM-A), which assesses symptoms of anxiety and their severity, widely used in both research and clinical settings. The scale consists of 14 items, each corresponding to a “category” of symptoms, either physical (for example, cardiovascular, gastrointestinal or respiratory symptoms) or mental symptoms (for example, tension, insomnia or anxious mood), with several examples of symptoms given in each category. Subjects were asked to assess to which extent they suffered from symptoms of the respective category and their severity. Each item was scored on a scale of 0 to 4 (0—“not present”; 4—“very severe”) [28]. The scale had excellent reliability in our sample, Cronbach’s alpha = 0.93.

### 2.4. Data Analysis

Data were analyzed using Rstudio and the script used for analysis is available in the Appendix A. First, we explored the percentage of missing values and normality assumption and we identified and excluded the multivariate outliers. The univariate normality assumption was examined via computing skewness and kurtosis, with values ranging between −2 and 2 being deemed acceptable [29]. The multivariate normality assumption was computed on the standardized residuals resulting from regressing all variables of interest on a random variable [30]. We computed the standardized residuals skewness and kurtosis as well. Furthermore, we also plotted the residuals as a histogram and Q-Q plot to visually explore their distribution. Multivariate outliers were identified using the Mahalanobis distance.

Inferential statistics were conducted using stepwise regression forward based on Akaike’s Information Criteria (AIC). In this method, each predictor is added based on the degree it reduces the value of AIC (the predictors that have the highest impact are added first). The process of adding more predictors is stopped when AIC no longer decreases [31].

## 3. Results

### 3.1. Descriptive Statistics

No missing values were observed. The univariate normality assumptions were met as skewness and kurtosis between −2 and 2 in all instances. Regarding multivariate normality, skewness and kurtosis were in the acceptable range for the standardized residuals, supporting the multivariate normality. Twenty-nine multivariate outliers (χ^2^ = 45.31, *p* < 0.001) were identified and excluded from the subsequent analyses. The mean, standard deviation, skewness and kurtosis for the variable of interest are presented in Table 1.

### 3.2. Stepwise Regression

#### 3.2.1. The Anxiety Level as Dependent Variable

The coping strategies included were focusing on and venting emotions (β = 0.30, *p* < 0.001), behavioral disengagement (β = 0.21, *p* < 0.001), acceptance (β = −0.10, *p* < 0.001), substance use (β = 0.16, *p* < 0.001), humor (β = −0.09, *p* < 0.001), positive reinterpretation and growth (β = −0.07, *p* < 0.010) and restraint (β = 0.05, *p* = 0.063). The total variance explained in anxiety symptoms was 25% (adjusted R2 = 0.25) (see Appendix A for more statistics).

#### 3.2.2. Well-being Measurements as Dependent Variables

Regarding physical health, the coping strategies included were behavioral disengagement (β = −0.22, *p* < 0.001), focus on and venting emotions (β = −0.24, *p* < 0.001), positive reinterpretation and growth (β = 0.19, *p* < 0.001), substance use (β = −0.11, *p* < 0.001), humor (β = 0.06, *p* < 0.015), denial (β = 0.07, *p* < 0.007), acceptance (β = 0.08, *p* < 0.004) and restraint (β = −0.05, *p* = 0.065). The total variance explained in physical health was 23% (adjusted R2 = 0.23) (see Appendix A for more statistics).

Regarding psychological health, the coping strategies included were positive reinterpretation and growth (β = 0.24, *p* < 0.001), focus on and venting emotions (β = −0.28, *p* < 0.001), behavioral disengagement (β = −0.26, *p* < 0.001), substance use (β = −0.15, *p* < 0.001), humor (β = 0.08, *p* < 0.001), use of emotional social support (β = 0.09, *p* < 0.001), denial (β = 0.08, *p* < 0.002), acceptance (β = 0.07, *p* < 0.009), restraint (β = −0.05, *p* = 0.063), religious coping (β = 0.04, *p* = 0.054) and planning (β = −0.04, *p* = 0.193). The total variance explained in physical health was 31% (adjusted R2 = 0.31) (see Appendix A for more statistics).

Regarding social relationships, the coping strategies included were positive reinterpretation and growth (β = 0.12, *p* < 0.001), behavioral disengagement (β = −0.26, *p* < 0.001), use of emotional social support (β = 0.27, *p* < 0.001), focus on and venting emotions (β = −0.22, *p* < 0.001), humor (β = 0.09, *p* < 0.001) and substance use (β = −0.06, *p* < 0.009). The total variance explained in physical health was 18% (adjusted R2 = 0.18) (see Appendix A for more statistics).

Regarding the environment, the coping strategies included were positive reinterpretation and growth (β = 0.16, *p* < 0.001), behavioral disengagement (β = −0.13, *p* < 0.001), focus on and venting emotions (β = −0.21, *p* < 0.001), use of emotional social support (β = 0.16, *p* < 0.001), substance use (β = −0.08, *p* < 0.001), religious coping (β = −0.07, *p* < 0.007), acceptance (β = 0.06, *p* < 0.019) and denial (β = −0.05, *p* = 0.078). The total variance explained in physical health was 15% (adjusted R2 = 0.15) (see Appendix A for more statistics).

## 4. Discussion

In a recent cross-sectional study by Maftei et al., 90 adolescents aged 11–15, residents of Iasi, Romania (i.e., 20,7 km from the Ukrainian border), took part in a self-reported assessment of peritraumatic exposure during the armed conflicts in Ukraine. The results confirmed a positive association between close war exposure, including helping behavior and a higher risk of peritraumatic dissociative experiences, anxiety symptoms, threat perception and lack of resilience [32]. Additionally, Mărcău et al. performed a similar investigation within the other, older age groups of Romanians. As such, 1193 subjects with a permanent residence in Romania and a minimum age of 18 years went to an online survey designed to assess the psychological effects associated with the fear of a close military conflict. Their results confirmed that the mental health and quality of life of people in Romania, as a state in the very close vicinity of Ukraine, are negatively affected by the fear that this war will escalate into a regional or global-scale conflict [33].

In a distinct study, Kostruba and Fishchuk surveyed 66 young Ukrainian volunteers. Their results show that media religious respondents tend to effectively use coping strategies, including problem analysis, recognition of one’s own worth, maintaining self-control and altruism, to deal with the psychological trauma imposed by the war [34]. Coping and resilience strategies among Ukraine war refugees have also been investigated by Oviedo et al. in an interview-based study on 94 refugees in different European countries (Poland, Italy and Spain) and 10 helping volunteers. Based on the obtained results, the authors could describe six clusters of coping mechanisms. “Relationships” was the strongest one (45%). The second most popular set of coping mechanisms was described as “Interior life” and gathered prayers, memories, beliefs, etc. The third cluster was labeled “Activity” and gathered some professional and recreational choices that proved effective in easing the traumatic pressure. Finally, the remaining clusters were “Therapy”, “Positive experiences” and “Good expectations”. These findings helped us to better understand the most prevalent coping mechanism among refugees of the Ukraine war and their relationship with achieving resilience [35].

Ilie et al. conducted a socio-economic assessment of the effects associated with the Russo—Ukrainian war on the Romanian population. Thus, 272 inhabitants from Craiova, Romania, completed the whole survey. The results indicated that up to 93% of the respondents are worried about the consequences of war and how it will affect them; 79.8% say they are afraid that the standard of living in Romania will suffer because of the war; up to 90% believe that the war will affect the Romanian trade, manufacturing and transport industry; 82.7% expect a possible economic crisis due to the SARS-CoV-2 pandemic followed by the Russo—Ukrainian war; 41.2% fear for the safety of their job; 40.4% are more careful with their spending; and 25% felt insecure lately. These socio-economic insecurities negatively impact the mental health of the Romanian population [17].

Based on the results obtained within this study, among the coping mechanisms investigated, focusing on and venting emotions (β = 0.30, *p* < 0.001) and behavioral disengagement (β = 0.21, *p* < 0.001) were the strongest predictors of anxiety. Focus on and venting emotions is the tendency to center the traumatic event and/or express the associated feelings. Hence, the negative effect of focusing on and venting emotions as a coping mechanism against anxiety did not come as a surprise. According to Marr et al., who coordinated a comprehensive investigation on 3294 adults, this coping method was noted as a potential longitudinal mediator between major depressive disorder (caused by different traumatic events) and generalized anxiety disorder [36]. A similar study by Liverant et al. among college students after the September 11th terrorist attacks found “focusing on and venting emotions” as an anxiety-predicting coping mechanism that, if performed excessively, can become maladaptive and negatively impact emotional adjustment when used to deal with traumatic stressors [37]. In fact, several other findings confirm the implications of focusing on and venting emotions in escalating post-traumatic emotions, such as anxiety, classifying it as a negative emotional coping method [24,27,38] or as a less useful coping strategy [39]. Likewise, behavioral disengagement, which is based on reducing one’s effort to deal with the stressor, was found to be consistently associated with anxiety, depression and overall poor mental health, thus being considered a reliable predictor for anxiety [40,41]. A growing body of evidence supporting the role of behavioral disengagement as a negative predictor for anxiety started to emerge on the basis of the COVID-19 pandemic and its associated emotional distress and trauma [42,43,44,45,46]. In 2016, Saxon et al. conducted a similar analysis, evaluating the coping strategies and mental health outcomes of conflict-affected persons in the Republic of Georgia. Their results pinpoint behavioral disengagement among the coping strategies significantly associated with PTSD symptoms and poor mental health, such as anxiety [47]. As such, behavioral disengagement is considered a negative coping strategy too [48,49]. Nonetheless, acceptance, which implies accommodating the situation as it is, appears to be the strongest negative predictor for war-associated anxiety. This was extensively confirmed by many other studies [50,51,52,53,54,55,56].

For physical health, behavioral disengagement and focusing on and venting emotions were the strongest predictors, both having a negative relation with physical health (among the investigated cohort, people who used these strategies more often reported poorer physical health). This negative association between behavioral disengagement and physical health was also confirmed by Boyraz et al., who assessed the preferred coping mechanism of 609 adults (18 years old or older) in relation to trauma and post-traumatic stress disorder [57]. Day and Livingstone made similar observations in a study conducted on the self-reported health symptoms of 521 military personnel [58]. Likewise, the negative association between focusing on and venting emotions as an avoidant coping strategy and physical health outcomes was also confirmed by Miola et al. in a study focusing on the emotional coping options during the second wave of the COVID-19 pandemic [46]. Meanwhile, the approach-oriented coping strategy called positive reinterpretation and growth, meaning reconstructing a stressful transaction in positive, bearable terms, predicted higher levels of physical health among the questioned Romanians during the Russo—Ukrainian war.

Similarly, positive reinterpretation and growth was the strongest predictor of psychological health (within the investigated cohort, people who used these strategies more often reported better overall psychological health). This was an expected result that further confirms and supports the findings of Litman and Lunsford who surveyed 450 individuals regarding the most traumatic event they have experienced in the past six months, where positive reinterpretation and growth were frequently and positively associated with psychological health and well-being [59]. This was also the case in the study by Cheshire et al., where the use of this coping method was examined among parents of children with cerebral palsy and positively correlated with self-efficacy and negatively correlated with depression and stress, thus being a predictor of physiological health [60,61]. Conversely, focusing on and venting emotions and behavioral disengagement had a negative impact on psychological health, as confirmed by other similar studies [62,63].

According to Hobfoll and London [64], social relationships are under a great deal of pressure in times of war, as conversations are misled by recurring rumors and interpersonal interactions are limited by war anxiety, while mental healthcare providers are also facing the same problems and, therefore, sometimes unable to provide adequate help [65]. Hence, when it comes to social relationships, the problem-focused coping method of positive reinterpretation and growth was the strongest positive predictor among our study cohort. In fact, the positive association between this coping strategy and meaningful social relationships contributes to a favorable outcome in terms of quality of life, while having a protective role against depression and anxiety [66]. In addition, the use of emotional social support, which is getting moral support, sympathy or understanding, was the second most reliable predictor for close social relationships, as expected based on their proven positive relationship [67,68]. Moreover, we observed and noted behavioral disengagement as a negative predictor for the quality of social relationships. However, this maladaptive coping strategy is known for its association with poor psychosocial outcomes and psychological health [69,70,71,72].

Finally, the positive emotional coping strategy of positive reinterpretation and growth was the strongest predictor of a perceived secure environment based on data collected from our respondents. This coping approach was also found to be positively associated with self-esteem, and individuals with higher self-esteem are prone to having a greater preference for problem-driven coping and proactive coping [35]. Altogether, based on the fact that, during our online survey, the war-associated effects, especially all the material damages, were kept outside the Romanian border, using the positive reinterpretation and growth method helped Romanians feel more secure in relation to their country, home and overall public environment. As expected, behavioral disengagement predicted a less perceived secure environment among the surveyed Romanians, which further confirms the negative effects of this maladaptive coping mechanism on the emotional status and overall patient well-being [73,74,75,76].

Overall, focusing on and venting emotions along with behavioral disengagement were the coping strategies that had the strongest negative impact on the anxiety and well-being of the Romanian population with respect to their perception of the nearby Russo—Ukrainian war. In contrast, positive reinterpretation and growth was the main coping strategy that predicted less anxiety and higher levels of well-being. This original, novel knowledge provides valuable resources for mental health practitioners that can be further translated into better care for Romanian patients that are emotionally overwhelmed by the war next door.

### Study Limitations

The study concept was based on the available literature limitations regarding the associative analysis between the coping mechanisms and the quality of life, respectively, and the mental health status of the Romanian citizens exposed to the Russo—Ukrainian war. However, our study encountered a few limitations of its own.

Our work is a type of observational research (cross-sectional) that analyzes data of variables collected at one given point in time across a predefined subset of the Romanian population. As such, the main weaknesses of this study include the inability to measure incidental parameters and to make a clear, accurate, causal inference. Additionally, this online survey did not assess the presence of any pre-existing mental disorders acutely manifested during the war or any psychiatric examinations or risk behavior.

Another limitation is the sample size, a rather small sizefor a study that aims to assess the coping mechanisms, quality of life and anxiety levels in response to the psychological trauma. Moreover, choosing such an accessible method as a Google Forms link to join the study leaves a large amount of bias as to who decides to fill out the survey. This assessment used self-reported measures; thus, result accuracy might be another important limitation. Finally, all analyses were modeled on a convenience sample of Romanian adults. Thus, the results presented in this paper are not to be considered representative of the entire Romanian population. The results should be cautiously interpreted considering the characteristics of the sample. In this context, the paper is relevant with respect to the relationship between psychological functioning and coping strategies in Romanians that have a college or a university degree, live in a city or town, are married and are predominantly women.

## 5. Conclusions

To the best of our knowledge, this online survey represents the first of its kind to ever be conducted in Romania. As such, we designed it to identify and assess the most frequently used coping mechanism among the Romanian population in response to their relationship with and direct implications of the Russo—Ukrainian war. Moreover, the associative relationship between these coping methods and perceived anxiety symptoms, physical health, psychological health, social relationships and environment was closely analyzed through the WHOQOL-BRE and COPE scales embedded in our online survey.

In this regard, our online, questionnaire-based assessment was focused on identifying the most frequently used coping mechanism among the Romanian population in response to their relationship with and direct implications of the Russo—Ukrainian war. Moreover, the associative relationship between these coping methods and perceived anxiety symptoms, physical health, psychological health, social relationships and environment was closely analyzed through the WHOQOL-BRE and COPE scales embedded in our online survey.

Based on the obtained results, among the Romanian population that engaged in our study (N = 1587), “focusing on and venting emotions “along with “behavioral disengagement” were the coping strategies that had the strongest negative impact on anxiety and well-being. These maladaptive, negative coping methods should raise concern among psychiatric medical doctors as they elevate the risk of severe anxiety and depression symptoms, which are closely correlated with poor life outcomes, including death by suicide. On the other hand, “positive reinterpretation and growth” was noted as the coping strategy that predicted significantly less anxiety, a higher degree of health and a better quality of life. Likewise, this approach-oriented emotional reaction should be taken into consideration as a positive coping mechanism useful when dealing with long-term, disturbing stressors, such as a nearby war.

These findings shed light upon the self-driven, psycho-regulatory mechanisms that are switched on among Romanians during this time of major crisis and provide valuable resources for mental health practitioners that can be further translated into a more efficient doctor—patient relationship. Therefore, we designed an accurate, cross-sectional assessment that facilitated the development of an extensive repertoire of reliable resources that can be successfully transferred into clinical practice to provide better care for all Romanian patients that are emotionally and mentally overwhelmed by the war next door. This may protect them against different psychological and physical health complaints.

## Figures and Tables

**Figure 1 healthcare-11-01412-f001:**
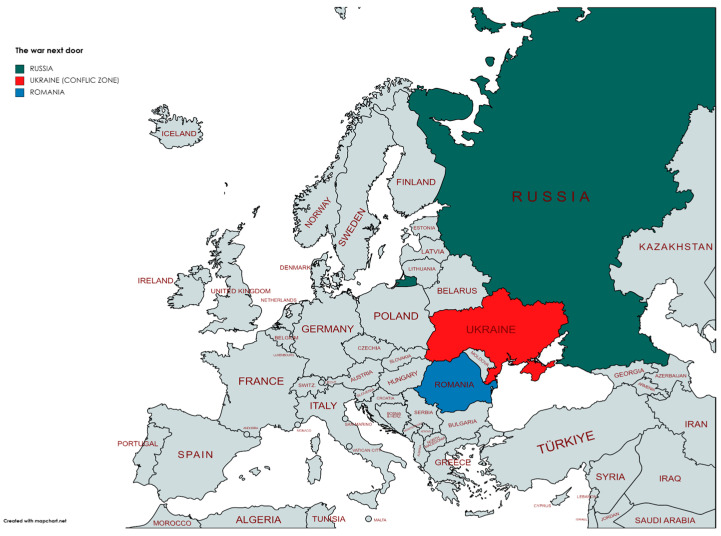
The war next door (generated with https://www.mapchart.net/index.html, accessed on 10 January 2023).

**Table 1 healthcare-11-01412-t001:** Descriptive statistics.

Variables	M	SD	Skewness	Kurtosis
Total anxiety	13.93	9.93	0.77	0.28
Positive reinterpretation and growth	11.73	2.26	−0.29	−0.15
Focus on and venting emotions	7.21	2.09	0.21	−0.31
Use of instrumental social support	10.78	2.67	−0.19	−0.25
Active coping	11.76	2.24	−0.15	−0.21
Denial	5.89	2.00	1.17	1.34
Religious coping	8.40	3.97	0.48	−1.03
Humor	7.86	3.40	0.60	−0.55
Behavioral disengagement	7.14	2.33	0.64	0.24
Restraint	9.97	2.18	−0.09	−0.11
Use of emotional social support	10.14	3.14	0.03	−0.72
Substance use	4.96	2.24	2.90	8.69
Acceptance	11.13	2.52	−0.26	−0.04
Suppression of competing activities	10.19	2.23	0.03	−0.02
Planning	12.35	2.50	−0.45	−0.15
Physical health	20.83	3.94	−0.40	0.08
Psychological health	20.11	3.90	−0.39	0.23
Social relationships	10.10	2.35	−0.46	0.03
Environment	27.76	4.15	−0.23	0.24

Note: M—mean, SD—standard deviation.

## Data Availability

A copy of the dataset resulting from the online survey (as an Excel sheet) is available in the Appendix A.

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
