# Peer review of "Coping Mechanisms during the War in Ukraine: A Cross-Sectional Assessment among Romanian Population"

_healthcare, 2023, doi:10.3390/healthcare11101412_

Round 1
Reviewer 1 Report
The manuscript appears to be well-written and provides important insights into the coping mechanisms of Romanians during the Russo-Ukrainian war. The authors provide a detailed description of their study methodology, including the measures used to assess quality of life and coping strategies. The findings are presented in a clear and concise manner, and the authors discuss the implications of their results for mental health practitioners.
One potential limitation of the study is that the sample was recruited through social media and email lists, which may have introduced selection bias. Additionally, the study was cross-sectional, which limits the ability to draw causal conclusions about the relationship between coping strategies and mental health outcomes. However, the authors acknowledge these limitations in their discussion section.
Overall, the manuscript provides valuable insights into the coping mechanisms of Romanians during the Russo-Ukrainian war and has the potential to inform mental health practice in this population.
The language is clear and accessible. I couldn't identify any misspellings or grammar mistakes.
Reviewer 2 Report
War and experiencing close military conflict hasn't been easy for the neighboring countries. The large refugee movements, air surveillance, nuclear war anxiety, and the negative economic impact have all contributed to the already upscaling fear of the war next door. This has increased the spectrum of emotional and cognitive distress, with a severe impact on the citizens' mental health, especially since the effects of this full-scale war were already visible in their everyday lives. It's very imporant issue, need to reserach it and focus on some kind of treatment and help.
It's not clear: The Russian-Ukrainian war that succeeded the COVID-19 pandemic continues to 46 deepen Europe's socio-economic fragility. - What do you mean?
Materials and methods - why there is such a big difference in the group of respondents among men and women?
Please explain in more detail the methodology and the questionnaire used.
Results is not clear, table isn't entirely clear, the description of the results is too sparse.
English is ok.
Reviewer 3 Report
You have mentioned the following keyword in your review: "attachment". Please ensure you have attached all necessary files.

Round 2
Reviewer 2 Report
Ok, thank yoy for all correct.